# Immunogenicity and Safety of the BNT162b2 mRNA COVID-19 Vaccine in Patients with Melanoma Treated with Immunotherapy

**DOI:** 10.3390/cancers14153791

**Published:** 2022-08-04

**Authors:** Panagiotis T. Diamantopoulos, Christina-Nefeli Kontandreopoulou, Aikaterini Gkoufa, Elena Solomou, Amalia Anastasopoulou, Eleni Palli, Panagiotis Kouzis, Spyros Bouros, Mihalis Samarkos, Gkikas Magiorkinis, Helen Gogas

**Affiliations:** 1First Department of Internal Medicine, Laikon General Hospital, National and Kapodistrian University of Athens, 11527 Athens, Greece; 2Department of Internal Medicine, University of Patras Medical School, 26500 Rion, Greece; 3Department of Hygiene, Epidemiology and Medical Statistics of the National and Kapodistrian University of Athens, 11527 Athens, Greece

**Keywords:** melanoma, immunotherapy, vaccination, immunogenicity, vaccine

## Abstract

**Simple Summary:**

The efficacy and safety of the BNT126b2 vaccine against SARS-CoV-2 has not been thoroughly studied in cancer patients treated with immunotherapy. This research aims to investigate the efficacy and safety of the vaccine in patients with melanoma under immunotherapy; at the same time, through the immunophenotyping of T cells and myeloid cells of the peripheral blood, it will be possible to look for changes in the subpopulations of such cells after vaccinations. The results of the study help establish the efficacy and safety of the vaccine in this population, especially since a theoretical concern exists about the vaccine triggering irAEs.

**Abstract:**

The BNT162b2 vaccine against SARS-CoV-2 has a proven efficacy and a favorable safety profile. In cancer patients under immunotherapy in the form of immune-checkpoint inhibitors (ICIs), the efficacy of the vaccine has not been thoroughly studied, while a theoretical concern has also been raised about triggering immune-related adverse events (irAEs) by the vaccine. We conducted a prospective, non-interventional study on the immunogenicity and safety of the BNT162b2 vaccine in patients with advanced or metastatic melanoma treated with ICIs. Blood samples were obtained 0–4 days before the first dose and 12–21 days after the second dose of the vaccine for the quantification of the SARS-CoV-2 anti-spike antibody using an ELISA and immunophenotyping of the T and myeloid cell subpopulations. The active recording of AEs for a two-month period was conducted. Forty patients were included in the study. All but one (97.3%) achieved seroconversion after two doses of the vaccine and no correlations of the antibody titers with any of the studied parameters (age, gender, stage and duration of the disease, type of ICI, previous treatment, etc.) were found. Moreover, no differences in the subpopulations of the T cells (including the T-regulatory cells) or the myeloid cells were found pre- and post-vaccination. All AEs were low-grade, while one case of arthritis exacerbation was noted. The seroconversion rate in the studied population was high and was comparable to that of healthy subjects, while no major safety issues were raised during the safety follow-up. Finally, no derangements in the subpopulations of T cells or myeloid cells were noted. This is the first study focusing on the immunogenicity, safety, and effect of anti-SARS-CoV-2 vaccines on the blood-cell immunophenotype status of patients with melanoma treated with ICIs.

## 1. Introduction

Cancer patients constitute a part of the general population with increased risks for severe Coronavirus Disease 2019 (COVID-19) symptoms, complications, and death [1,2]. Although in the beginning of the pandemic, cancer management, and treatment strategies were debated as possible risk factors of a dismal course of the disease, Ref. [3] a recent systematic review reported that receiving chemotherapy, radiation therapy, hormonal therapy, or immunotherapy is not associated with an increased risk for adverse outcomes [4].

Due to the high contagiousness of the virus and increased mortality rates, primary prevention became a global priority, with active immunization being a principal goal. Many studies from different countries have reported high efficacy and good tolerability for mRNA vaccines against SARS-CoV-2 [5]. The BNT162b2 vaccine against SARS-CoV-2 induces highly effective humoral and cell-mediated immune responses, with the production of neutralizing-IgG antibodies as well as the release of cytokines, such as interferon gamma (IFNγ), tumor necrosis factor alpha (TNFa), interleukin (IL)-2 and IL-12p70, and the activation of virus-specific CD4^+^ and CD8^+^ T cells [6,7,8].

However, data concerning COVID-19 vaccine efficacy and safety in patients with an underlying malignancy being treated with immunotherapy are not mature and studies exclusively designed for this group are scarce [9,10]. The aim of this study is to evaluate the immunogenicity and safety of the BNT162b2 vaccine in patients with advanced or metastatic melanoma treated with immune-checkpoint inhibitors (ICIs). 

## 2. Patients and Methods

### 2.1. Patients

Adult patients with melanoma under treatment with an ICI or a combination of ICIs were informed about the study and participated after providing written informed consent. Patients who were willing to be vaccinated against SARS-CoV-2 with the BNT162b2 mRNA COVID-19 vaccine according to the national vaccination program were included in the study. Exclusion criteria included vaccination with other vaccines against SARS-CoV-2, known human immunodeficiency virus infections, and the inability to provide written informed consent. Moreover, patients with a known history of COVID-19 before the administration of the first dose of the vaccine were excluded from the immunogenicity analysis. The baseline epidemiological, clinical, and laboratory characteristics of the patients as well as the treatment data were recorded. The age and disease stage at the time of vaccination, disease duration, as well as data on the treatment of patients (treatment lines, previous treatments, type of ICI, duration of treatment with ICIs, response to treatment) were also collected and analyzed. All patients underwent treatment with at least one ICI at the time of vaccination.

### 2.2. Vaccination

Patients were vaccinated with two 30 mcg doses of the BNT162b2 mRNA COVID-19 vaccine, administered intramuscularly in the deltoid muscle 21 days apart, according to the national program for vaccination against SARS-CoV-2. 

### 2.3. Study Procedures

The study was designed to assess immunogenicity at a baseline (0–4 days before the first dose of the vaccine) and within 12–21 days of the second dose of the vaccine. Blood samples were collected at predefined time points following a standard venipuncture procedure. 

#### 2.3.1. Immunogenicity—Antibody Testing

Sera were obtained after centrifugation, aliquoted in ice, and stored at −80 °C until use. The sera were then tested using an enzyme-linked immunosorbent assay (ELISA) for the presence of IgG antibodies against the SARS-CoV-2 spike protein as described below.

The Wantai SARS-CoV-2 IgG ELISA (Beijing Wantai Biological Pharmacy Enterprise, Beijing, China), intended for the quantitative detection of IgG-class antibodies to SARS-CoV-2 in human sera or plasma, was run according to the manufacturer-provided protocol for the Elisys Uno (Human Diagnostics, Wiesbaden, Germany) automated instrument. The assay is based on an indirect ELISA principle that detects IgG antibodies binding the SARS-CoV-2 receptor binding domain (RBD) of the S1 subunit of the spike protein. Briefly, 10 μL-serum samples were added to SARS-CoV-2 recombinant antigen-pre-coated microplate wells and incubated at 37 °C for 30 min. Wells were washed, and the horseradish peroxidase (HRP)-conjugated SARS-CoV-2 antigen was added. After incubation at 37 °C for 30 min, wells were washed and two chromogen solutions were added. Following 15 min of incubation at 37 °C, the reaction was halted using a “stop solution” and absorbance was measured on the dual-filter instrument at 450 nm after setting the reference wavelength at 600–650 nm.

The lower limit of detection for the reaction was 10 AU/mL. The clinical sensitivity was 94.94% (95% confidence interval (CI), 87.69–98.01%) in samples collected after ≥15 days from the onset of symptoms from patients who were confirmed as positive SARS-CoV-2 cases by a real-time polymerase chain reaction at a cutoff value of 10 AU/mL. In the present study, the cutoff point of 10 AU/mL was used to define seroconversion. 

#### 2.3.2. T-Lymphocytes and Neutrophil Immunophenotyping

Peripheral blood mononuclear cells (PBMCs) were obtained by density-gradient centrifugation, as previously described [11]. The percentages and absolute numbers of each population of interest were stained with anti-CD4, CD8, CD25, and PD1 monoclonal antibodies for T cells. The staining panels of antibodies designed to facilitate fluorescence-activated cell sorting (FACS) were developed to sort isolated PBMCs into five different lymphoid subtypes (CD4^+^CD25^+/−^ T cells, CD4^+^CD25highFoxp3^+^ T cells representing the T-regulatory cell compartment, CD8^+^ T cells, and PD1^+^ cells). Additionally, seven myeloid subtypes of cells were examined based on the percentages and absolute numbers of the following subpopulations: HLADR^+^CD14^+^CD16^+/−^ cells, HLADR^+^CD14^−^CD16^+^, HLADR^+^CD33^+/−^, HLADR^intermediate^CD33^+^, and PDL1^+^ cells. The preparation of the cells and staining were performed as previously described [12]. Table 1 provides details for each of the antibody/fluorophore conjugates used in this panel. Analyses were performed using the BD-FACSAria cell sorter (BD Biosciences, Franklin Lakes, NJ, USA) and FlowJo, v10 software [13]. 

#### 2.3.3. Safety Follow-Up

Local or systemic adverse events (AEs) within seven days of each dose of the vaccine were actively recorded. In addition, patients were followed for the following two months for late AEs. The AEs were captured during the post-vaccination sample-collection visit and during a phone call or visit two months after the second dose. Specific questions about local (pain or edema) or systematic (fever, malaise, and headache) adverse events as well as the use of antipyretic or analgesic medication during the first week after each dose were posed to the patients.

This was a single-center non-interventional prospective study approved by the Institutional Review Board of the participating center (IRB protocol number 67/25.01.21). All experiments were performed in accordance with the Declaration of Helsinki.

#### 2.3.4. Statistical Analysis

Statistical analyses were conducted using the IBM SPSS Statistics software, version 26 (IBM Corporation, North Castle, NY, USA). Correlations between categorical variables were tested using the Pearson’s Chi-Square test or a Fisher’s exact test for analyses of categories with expected values below five; the Independent-Samples Mann–Whitney U test was used for testing between a categorical variable with two levels and not normally distributed continuous variables; the Kruskal–Wallis H test was used for categorical variables with more than two levels. A paired *t*-test was run to assess associations between the blood cell populations before and after the vaccination. The level of significance for all statistical tests was set at a probability value of lower than 5% (2-sided *p* < 0.05).

## 3. Results

The study comprised 40 patients with melanoma treated with immunotherapy at the time of vaccination. All patients received two doses of the BNT162b2 mRNA COVID-19 vaccine 21 days apart. The pre-vaccination sample was taken at a median time of 2 (0–4) days before the first dose of the vaccine. The post-vaccination sample was taken at a median time of 14 (13–17) days after the second dose of the vaccine. Three (7.5%) patients reported a positive test (rapid-antigen test or PCR) for SARS-CoV-2 during the six months before vaccination, and the pre-vaccination immunogenicity status was positive in all three of them. These patients were excluded from the immunogenicity/seroconversion analysis.

### 3.1. Immunogenicity/Seroconversion Results

All but one (36/37, 97.3%) patient achieved seroconversion post-vaccination, with a median antibody titre of 28.47 AU/mL (90% Confidence Interval: 10.94–33.69). The antibody titre did not correlate with any of the studied variables (i.e., age, gender, melanoma stage, disease duration, previous treatment lines, immunotherapy type, treatment duration, emergence of irAEs during immunotherapy, and AEs attributable to the vaccination), and there was a non-statistically significant trend for lower antibody titres in patients actively treated with corticosteroids (N = 2) for irAEs (16.59 AU/mL vs. 28.96 AU/mL, *p* = 0.123). The main characteristics of the studied populations as well as the immunogenicity results are summarized in Table 2. 

### 3.2. Immunophenotype Results

Analyses of the numbers of CD4^+^CD25^+^ cells did not reveal any statistically significant differences before and after vaccination (mean ± standard error of the mean (SEM), 7.06% ± 0.83 vs. 5.66% ± 0.45; *p* = 0.13). The percentages of CD4^+^CD25^hi+^ cells were comparable before and after vaccination (mean ± SEM, 1.61% ± 1.4 vs. 1.25% ± 0.75; *p* = 0.68), while the Foxp3^+^ subpopulation of the CD4^+^CD25^hi+^ cells, representing the regulatory-T cell compartment (Tregs), did not revealed any variances before and after vaccination (mean ± SEM, 46.08% ± 3.78 vs. 47.12% ± 4.74; *p* = 0.86). The same applied for CD8^+^ cells (mean ± SEM, 18.44% ± 2.38 vs. 14.78% ± 2.32, *p* = 0.49).

Similar results were obtained when the HLA-DR^+^CD14^+^CD16^−^, HLA-DR^+^CD14^+^CD16^+^, and HLA-DR^+^CD14CD16^+^ cell populations were examined (mean ± SEM, 13.94% ± 2.74 vs. 13.14% ± 2.56; *p* = 0.99–26.11% ± 2.73 vs. 30.48% ± 2.54; *p* = 0.35–15.6% ± 1.85 vs. 16.17% ± 2.19; *p* = 0.84, respectively). We further analyzed the percentages of HLA-DR^+^CD33^+^ and CD33^+^ populations along with the CD33^+^HLA-DR^intermediate^ populations and found no statistically significant differences in any of those three populations before and after vaccination (mean ± SEM, 49.28% ± 6.45 vs. 39.18% ± 3.56; *p* = 0.83–26.11% ± 2.73 vs. 30.48% ± 2.54; *p* = 0.35–15.6% ± 1.85 vs. 16.17% ± 2.19; *p* = 0.84, respectively).

When we analyzed the mean fluorescence intensity (MFI) of PD1 on CD8^+^ cells and CD4^+^CD25^−^ cells, there was no significant difference before and after vaccination (mean ± SEM, 159.6 ± 11.4 vs. 166.8 ± 11.37; *p* = 0.56 and 44.45 ± 6.23 vs. 34.76 ± 3.41; *p* = 0.56, respectively). Of interest, the MFI of PD1 on CD4^+^CD25^+^ cells was significantly higher before vaccination compared to the values recorded after vaccination (mean ± SEM, 139.2 ± 13.56 vs. 106 ± 4.93, *p* = 0.017, Figure 1). The MFI of PDL1 did not differ in any of the analyzed populations. All immunophenotype results are presented in Table 3.

### 3.3. Safety Results

Adverse events (AEs) during the two-month safety follow-up period were reported in 17 patients; 10 (25.0%) patients experienced an AE after the first dose and 15 (37.5%) patients experienced an AE after the second dose of the vaccine. The majority of AEs were either local (4/10 and 6/15 patients after the first and second dose, respectively) or low grade (1 or 2) systemic ones (fever, 2/10 and 10/15; malaise, 3/10 and 10/15; headache, 1/10 and 0/15 after the first and second dose, respectively). One patient presented three days after the first dose with a deterioration of a previous immune-related small-joint arthritis that was effectively treated with low-dose corticosteroids, while one patient was diagnosed with a pulmonary embolism three weeks after the second dose. The patient was a man at his early eighties with long-standing stage-IV melanoma in complete remission under pembrolizumab during the preceding 26 months. He was asymptomatic and the pulmonary embolism was found after a scheduled restaging chest computed-tomography scan. 

The emergence of AEs (local or systematic) did not correlate with the post-vaccination antibody titre. Moreover, it did not correlate with gender, the stage of melanoma, or the type of immunotherapy. AEs were more prevalent in younger patients (the patients reporting AEs had a median age of 55.5 years vs. 69.0 years for patients not reporting AEs, *p* = 0.017), patients with long-standing disease (43.4 months vs. 32.9 months, respectively, *p* = 0.046), and finally, patients with a longer duration of immunotherapy (19.2 months vs. 5.1 months, respectively, *p* = 0.020). A logistic regression was performed to ascertain the effects of age, gender, disease duration and stage, treatment lines, and immunotherapy type and duration on the likelihood that the patients have an AE. Only an increasing duration of immunotherapy was associated with an increased likelihood of experiencing an AE (odds ratio 1.199, *p* = 0.047). 

## 4. Discussion

Protecting cancer patients from vaccine-preventable infections is an essential part of disease management; further, non-live vaccines are safe in patients receiving chemotherapy and/or radiation therapy, as well as in transplant recipients. Nevertheless, vaccine efficacy is usually reduced due to disease- and treatment-related immunosuppression [14] Moreover, in patients receiving immunotherapy in the form of ICIs, there is a theoretical concern about vaccines triggering irAEs, and data on the efficacy and safety of vaccinations against infectious agents in this population are scarce. The use of mRNA vaccines against SARS-CoV-2 has further increased uncertainty about the efficacy of this new class of vaccines as well as safety related to the emergence of irAEs.

Several studies have evaluated the use of mRNA vaccines in patients treated with ICIs, and initial results showed an efficacy of the BNT162b2 vaccine in cancer patients treated with ICIs comparable to that of the general population in terms of both humoral and cell immune responses, while irAE rates were low [10,15,16,17].

In the present study, a very high seroconversion rate (97.3%) was found among patients with melanoma treated with ICIs. Although preliminary reports show that neutralizing antibodies against SARS-CoV-2 were significantly lower in patients compared to matched healthy volunteers [15], recent studies have reported comparably high (95.0–97.0%) seroconversion rates among cancer patients treated with ICIs that were significantly higher than those of cancer patients treated with chemotherapy or targeted therapy [18,19]. Vaccine efficacy has been also studied for influenza vaccines and has been found to be higher in patients treated with ICIs in comparison to chemotherapy [20,21].

Moreover, no correlation was found between the seroconversion rate or the antibody titre and age, gender, stage and the duration of the disease, or the number of previously administered treatment lines. A non-statistically significant trend for lower antibody titres in patients treated with corticosteroids for previously diagnosed irAEs was noted, but the number of patients on corticosteroids was small; thus, no solid conclusions can be drawn. It is very interesting that factors associated with low seroconversion rates and low antibody titres, such as older age, stage of the disease, and the duration of treatment, in studies on cancer patients treated with other treatment types such as chemotherapy or monoclonal antibodies and targeted therapy [19] did not seem to affect immunogenicity in patients treated with ICIs. It should be noted although that these factors have not been thoroughly studied in patients treated with ICIs, since there are only a handful of studies available and most of them are not focused on patients treated with ICIs. Nevertheless, since ICIs are considered non-immunosuppressive, these results are somewhat expected.

As regards to safety, AEs were generally mild and transient and occurred more frequently after the second dose of the vaccine. AEs were more prevalent in younger patients, patients with long-standing disease, and patients who underwent a longer duration of immunotherapy. The correlation of the emergence of AEs with a younger age has been described in the past, although older adults tended to report more serious AEs [22,23]. To the best of our knowledge, a higher prevalence of AEs in patients with long-standing disease or long duration of immunotherapy has not been reported, but this may reflect a more robust reaction of the immune system and cytokine production in patients with effectively activated T-cells. Cytokine-release syndrome-like serum responses have been recently reported in cancer patients under immunotherapy who have been vaccinated against SARS-CoV-2 [24,25].

The vaccine-related AE prevalence was not correlated with the immunogenicity of the vaccine either. It is well-known that the emergence of irAEs in cancer patients treated with ICIs is correlated with a favorable prognosis, possibly indicating cross-reactivity between anti-tumor and anti-self-immune reactions [26] or a higher level of T-cell activation by the ICIs, possibly leading to more durable responses. Thus, although it has been shown that in patients with a hemato-oncological disease, vaccine-related AEs are more common in seroconverted patients after vaccination [27], this was not confirmed by our results. 

Whether the incidence rate of irAEs increases after COVID-19 vaccines warrants further investigation. Some evidence for the safety profile of vaccination in cancer patients could be provided by studies examining the immunogenicity of influenza vaccination in patients receiving ICI therapy, which reveal comparable results to that of healthy individuals, while the risk of irAEs was unsubstantial [21,28]. Another study investigated influenza-specific immune responses, as well as the risk of irAEs after vaccination in lung cancer patients under PD1 blockade and concluded that humoral-immune responses were similar between cancer patients and healthy controls; however, the risk for development of irAEs was higher in patients under checkpoint inhibitor therapy. [29] A correlation of irAEs and vaccination could be attributed to the increased cytokine production after anti-CTLA4 and anti-PD1/PDL1 administration, as both resulted in enhanced CD4^+^ and CD8^+^ T-cell activation with a subsequent release of cytokines such as TNF, IFNγ, and IL-2 [30].

Finally, immunophenotyping of the T and myeloid cells failed to show any statistically significant differences before and after vaccination. A possible explanation could be that the vaccination itself cannot perturb the relations in those populations established by the use of ICIs. The single most important finding was a significantly higher MFI of PD1 on CD4^+^CD25^+^ cells before vaccination vs. after vaccination; however, this is difficult to interpret and warrants further investigation. Although it is difficult to decipher this alteration and the exact mechanism cannot be determined, we suspect that it represents a stabilization mechanism that CD4^+^CD25^+^ T cells use in patients with melanoma; the PD1/PDL1 axis is used by regulatory-T cells to suppress autoreactive-B cells in vivo [31]. Thus, we can assume that this upregulation is the result of B-cell activation and antibody production after vaccination in order to suppress the development of autoreactive B cells.

The strengths of the present study include a well-characterized and homogeneous population of patients with melanoma treated with ICIs and vaccinated with only one type of mRNA vaccine and the attempt to find possible effects of the vaccination on the subpopulations of T cells and myeloid cells. The main limitation of the study is the rather small number of participants and the fact that the almost-complete response of the patients to the vaccine prevented any statistical analyses on factors affecting seroconversion. Nevertheless, it should be noted that this is the only available study so far that provides focused results on the immunogenicity and safety of the BNT162b2 vaccine in patients with melanoma under ICI.

## 5. Conclusions

The results of the present study show that the BNT162b2 vaccine against SARS-CoV-2 is effective and safe in patients with melanoma treated with ICIs. Moreover, no significant effects on the subpopulations of T cells and myeloid cells were noted.

## Figures and Tables

**Figure 1 cancers-14-03791-f001:**
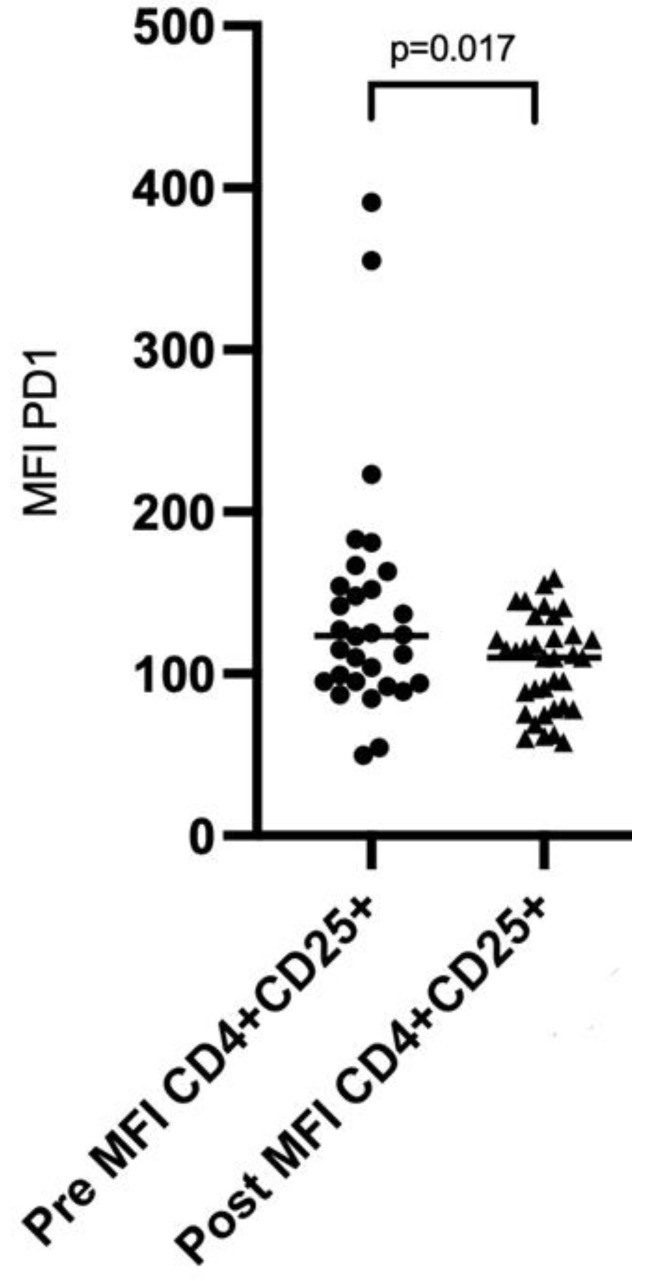
Scatter plot showing the distribution of the MFI of PD1 on CD4^+^CD25^+^ before and after vaccination.

**Table 1 cancers-14-03791-t001:** Antibodies used for flowcytometry.

Marker	Fluorophore	Manufacturer/Cat Number	Clone
CD4	BV510	Biolegend/317444	OKT4
CD8	APC-Cy7	Biolegend/344714	SK1
CD25	FITC	Biolegend/356106	M-A251
FOXP3	BV421	Biolegend/320124	206D
PD1	PeCy7	Biolegend/329918	EH12.2H7
HLADR	APC	Biolegend/307610	L243
CD14	FITC	Biolegend/555397	M5E2
CD16	PeCy7	Biolegend/302016	3G8
CD33	PE	Biolegend/366608	P67.6
PDL1	PerCp-Cy5.5	Biolegend/329738	29E.2A3

**Table 2 cancers-14-03791-t002:** Patient characteristics.

Characteristic	Result
Number of patients, N (%)	40 (100)
Male/female, N (%)	25/15 (62.5/37.5)
Age (years), Median (range)	66.0 (40.0–84.0)
Melanoma stage at vaccination, N (%)	
II	1 (2.5)
III	17 (42.5)
IV	22 (55.0)
Disease duration (at vaccination), (months), Median (range)	34.8 (3.1–220.1)
Treatment lines (including current), Median (range)	
1	18 (45.0)
2	5 (12.5)
3	7 (17.5)
>3	10 (25.0)
Immunotherapy type, N (%)	
CTLA4-inhibitor	0 (0)
PD1-inhibitor	29 (72.5)
PDL1-inhibitor	5 (12.5)
Combined CTLA4 and PD1-inhibitor	6 (15.0)
Duration of immunotherapy (months), Median (range)	6.6 (0.6–48.9)
Cycles of treatment, Median (range)	8.0 (1.0–49.0)
Treatment with corticosteroids at vaccination, N (%)	2 (5.0)
Time interval (2nd dose to blood sampling) (days), Median (range)	14 (14–17)
Patients with adverse events (First dose), N (%)	10 (25.0)
Patients with adverse events (Second dose), N (%)	15 (37.5)
Antibody titre (pre-vaccination), Median (range)	0.01 (0.00–30.78)
Immunogenicity (pre-vaccination), N (%)	3 (7.5)
Antibody titre (post-vaccination), Median (range)	28.47 (8.49–34.46)
Immunogenicity (seroconversion/post-vaccination), N (%)	36/37 (97.3)

**Table 3 cancers-14-03791-t003:** Immunophenotype results of blood cell subpopulations before and after vaccination with the BNT162b2 vaccine against SARS-CoV-2.

Cell Subpopulation	Pre-Vaccination Result (Mean ± SEM)	Post-Vaccination Result (Mean ± SEM)	Statistical Significance (2-Sided *p*)
CD4^+^CD25^+^	7.06% ± 0.83	5.66% ± 0.45	0.13
CD4^+^CD25^hi+^ *	1.61% ± 1.41	1.25% ± 0.75	0.68
CD4^+^CD25^hi+^Foxp3^+^	46.08% ± 3.78	47.12% ± 4.74	0.86
CD8^+^	18.44% ± 2.38	14.78% ± 2.32	0.49
HLA-DR^+^CD14^+^CD16^−^ *	13.94% ± 2.74	13.14% ± 2.56	0.99
HLA-DR^+^CD14^+^CD16^+^	26.11% ± 2.73	30.48% ± 2.54	0.35
HLA-DR^+^CD14CD16^+^	15.6% ± 1.85	16.17% ± 2.19	0.84
HLA-DR^+^CD33^+^	49.28% ± 6.45	39.18% ± 3.56	0.83
CD33^+^	26.11% ± 2.73	30.48% ± 2.54	0.35
CD33^+^HLA-DR^intermediate^	15.6% ± 1.85	16.17% ± 2.19	0.84
MFI of PD1 on CD8^+^	159.6 ± 11.4	166.8 ± 11.37	0.56
MFI of PD1 on CD4^+^CD25	44.45 ± 6.23	34.76 ± 3.41	0.56
MFI of PD1 on CD4^+^CD25^+^	139.2 ± 13.56	106 ± 4.93	0.017

SEM, standard error of the mean; MFI, mean fluorescence intensity. * In case of non-normal distribution of the values, a non-parametric test (Related-Samples Wilcoxon-Signed Rank Test) was used. In all other (normally distributed) groups of values, a paired-samples *t*-test was used.

## Data Availability

Data on the experiments as well as patient data are available upon reasonable request. For original data, please contact pandiamantopoulos@gmail.com.

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
