# Peer review of "Immunogenicity and Safety of the BNT162b2 mRNA COVID-19 Vaccine in Patients with Melanoma Treated with Immunotherapy"

_cancers, 2022, doi:10.3390/cancers14153791_

Round 1

Reviewer 1 Report

In this manuscript, the authors report the results of a small study (N=40, but effectively 37) immunogenicity, safety, and effect on the blood cell immunophenotype status of anti-SARS-Cov-2 vaccines in patients with melanoma treated with ICIs.  Statistical analyses were performed to compare pre- to post- vaccination proportions of various immune cell types and evaluate the rate of Adverse events (AEs) as dependent on clinicopathologic and treatment factors. The statistical analyses and reporting of the results need to be improved as following detailed below.

1.     The results of paired t-test are reported for differences between the blood cell populations before and after the vaccination but the normal distribution assumption is not validated.

2.     For reporting the rate of seroconversion post-vaccination (36/37=93%), it is necessary to include the confidence interval for this proportion.

3.     Analysis of AEs is limited to univariate comparison of the patient characteristics between groups with and without AEs. A multivariable model for the rate of AEs as dependent on patient characteristics should be considered. The AEs after the first and the second dose may be combined for higher power and analysis has to account for correlation due to repeated observations per patient.

4.     The authors may consider reporting the results for blood cell populations before and after the vaccination as a table for better presentation.

Author Response

File uploaded as a word document.

Reviewer 2 Report

In this study, Diamantopoulos et al. aimed to evaluate the efficacy and safety of the BNT126b2 vaccine against SARS-CoV-2 in melanoma patients treated with immune checkpoint inhibitors. The study is well designed and the data is clearly presented. However, I have several suggestions which could improve the study:

11.  In the Introduction section, lines 57-59 – you can cite another study regarding the efficacy and safety of the vaccine against SARS-CoV-2 in cancer patients receiving immunotherapy. See Strobel et al., 2021(doi: 10.1007/s00262-021-03133-w),

22.    There is no information about the control group patients(sera) used in the study. 

33.    It would be interesting to investigate the cytokine release syndrome-like serum responses in melanoma patients treated with immune check point inhibitors as an adverse effect post-vaccination; see Walle et al.,2022(doi.org/10.1038/s43018-022-00398-7),Au,L.et al.2021(doi.org/10.1038/s41591-021-01387-6)

Author Response

File uploaded as a word document.

Round 2

Reviewer 2 Report

The authors addressed all the questions/suggestions.